# Bio-Based Alkali Lignin Cooperative Systems for Improving the Flame Retardant and Mechanical Properties of Rigid Polyurethane Foam

**DOI:** 10.3390/polym15244709

**Published:** 2023-12-14

**Authors:** Xu Li, Chang Liu, Xinyu An, Li Niu, Jacko Feng, Zhiming Liu

**Affiliations:** 1College of Material Science and Engineering, Northeast Forestry University, Harbin 150040, China; lixu@nefu.edu.cn (X.L.); liuchang1640@163.com (C.L.); anxinyu1997@163.com (X.A.); nefuniuli@126.com (L.N.); 2Aulin College, Northeast Forestry University, Harbin 150040, China; jacko.feng@aemg.com.au

**Keywords:** alkali lignin, rigid polyurethane foam, flame retardancy, mechanical properties

## Abstract

Lignin was utilized as an environmentally friendly synergistic agent to augment the fire resistance and mechanical characteristics of rigid polyurethane foam (PUF)/melamine–formaldehyde resin ammonium polyphosphate (MFAPP). The incorporation of lignin significantly enhanced the charring capability and flame retardancy of PUF/MFAPP. Specifically, PUF/MFAPP_12_/A-lignin_3_ exhibited a charring residue of 23.1% at 800 °C, accompanied by an increase in the limiting oxygen index (LOI) to 23.1%, resulting in a UL-94 V-0 rating. The cone calorimeter test (CCT) revealed that the peak heat release rate (PHRR), total heat release (THR), smoke production rate (SPR), and total smoke production (TSP) values of PUF/MFAPP_12_/A-lignin_3_ were all lower than for pure PUF. MFAPP and alkali lignin exerted a noticeable influence on the physical and mechanical properties, leading to increases in density (35.4 kg/m^3^), thermal conductivity (32.68 mW/(m·K)), and compressive strength (160.5 kPa). Observations of the morphology and elemental composition of char residues after combustion indicated the formation of an intact, thick, and continuous char layer enriched with nitrogen and phosphorus elements, which acted as a protective shield for the underlying foam.

## 1. Introduction

Rigid polyurethane foam (PUF) has found extensive applications across various sectors, including aviation materials, construction materials, electronics, electrical appliances, transportation, and petroleum pipelines, thanks to its outstanding physical and mechanical properties, insulation capabilities, and resistance to aging [1,2,3,4,5]. However, PUF is inherently highly flammable, characterized by a low limiting oxygen index (LOI) of approximately 18%. In the event of a fire, PUF exhibits rapid combustion with swift flame propagation and the release of substantial heat. Simultaneously, the combustion process generates toxic gases like hydrogen cyanide (HCN) and carbon monoxide (CO), posing significant threats to both human health and environmental safety [6,7,8,9]. Consequently, it becomes imperative to enhance the fire safety properties of PUF.

In recent years, scientists have dedicated their efforts to exploring a range of flame retardants (FRs) aimed at enhancing the fire resistance of rigid polyurethane foam [10,11,12]. Several kinds of phosphorus-containing FRs were used as halogen-free FRs for flame-retardant PUF. Dimethylmethylphosphonate (DMMP) [13], ammonium polyphosphate (APP) [14], and DOPO [15] were studied in terms of both flame retardancy and the compressive performance of modified PUFs. Among these options, APP has garnered significant attention due to its superior phosphorus–nitrogen ratio and its exceptional flame-retardant properties when applied to PUF [14,16]. In intumescent flame-retardant systems, APP consistently serves as both an acid and gas source, producing phosphoric acid, polyphosphoric acid, and non-combustible gases like ammonia [17,18]. Although APP endows PUF with higher flame retardancy, it still has some defects, such as easy moisture absorption and poor compatibility with polymers. Some researchers have shown that modification of APP by coating it with microcapsules of melamine–formaldehyde resin (MFAPP) can effectively reduce the solubility of APP in water and improve its compatibility with the polymer substrate and its flame retardancy [19,20,21].

Simultaneously, as science, technology, and society undergo rapid advancements, fossil petroleum-based resources are progressively being depleted [22,23]. This depletion has significantly constrained the versatile applications of various petroleum-based polymer monomers, leading to growing attention to biomass materials [24]. Among these, lignin has emerged as a prominent candidate, ranking second only to cellulose in terms of abundance [25]. Due to the substantial presence of alcohol hydroxyl [26,27,28], phenol hydroxyl [29,30], carbonyl, methoxyl [31,32], and carboxyl groups within its structural units, lignin has found widespread utility in the synthesis and modification of materials such as polyurethane, phenolic resin, epoxy resin, and others. Among these applications, the development of flame-retardant polyurethane foams utilizing lignin and its derivatives has become a focal point of research. For example, a flame-retardant modified rigid polyurethane foam was synthesized by replacing some of the polyols with hydroxymethylated lignin and using hybrid silicon as a flame retardant. Study results showed that the hybrid silicon improved the thermal stability of rigid polyurethane foam and increased the amount of char residue remaining after decomposition of the foam [33]. Furthermore, three types of lignin, including sodium lignosulfonate (LS), alkaline lignin (AL), and enzymatic hydrolysis lignin (EHL), were individually applied to significantly enhance the flame retardance and thermostability of PU foams [34]. In summary, the development of lignin-based flame-retardant rigid polyurethane foam materials holds immense significance and practical value.

In this work, the flame retardancy of MFAPP with refined alkali lignin in PUF was researched in detail. This investigation delved into the impact of alkali lignin on several key aspects, including the thermal stability, flame resistance, combustion characteristics, thermal conductivity, storage modulus, and mechanical attributes, of the PUF/MFAPP composite. Finally, the flame-retardant mode of action was revealed in detail.

## 2. Experimental Section

### 2.1. Materials

Polyether polyol (4110A) (hydroxyl number, 450 mg KOH/g; viscosity, 2500–3000 mPa·s, at 25 °C) was purchased from Harbin Feiyue Insulation Material Co., Ltd. (Harbin, China). Polyaryl polymethylene isocyanate (PAPI), PM-200 (viscosity at 25 °C, 150–250 mPa·s; -NCO content 30.5–32 wt%) was obtained from Yantai Wanhua Polyurethane Synthetic Materials Co., Ltd. (Yantai, China). Silicone oil (F-8805) was provided by Shandong Guolan New Material Co., Ltd. (Dongying, China). Dibutyltin dilaurate (DBTDL) was bought from Shanghai Aladdin Reagent Co. Ltd. (Shanghai, China). Industrial alkali lignin was purchased from Shenyang Puhe Chemical Co., Ltd. (Shenyang, China). Ammonium polyphosphate coated with melamine–formaldehyde resin (MFAPP) was supplied by Guangzhou Yinyuan New Materials Co., Ltd. (Guangzhou, China). Sodium hydroxide (NaOH, 96%) and concentrated hydrogen chloride (HCl, 36%) were obtained from Tianjin Tianli Chemical Reagents Co., Ltd. (Tianjin, China). The deionized water was laboratory-made.

### 2.2. Refinement of Alkali Lignin (A-Lignin)

Due to a large number of impurities in the industrial alkali lignin, it needed to be refined before use. The industrial alkali lignin was adjusted to a mass fraction of 30% with distilled water, and by using 10 wt% NaOH solution, we adjusted the pH to 13–14 with 10 wt% NaOH solution to completely dissolve the alkali lignin. The mixture solution was filtered to remove the insoluble material, and then the pH was adjusted to 2 with 12% hydrochloric acid at 60 °C so that the alkali lignin was completely precipitated. The resulting brown precipitate was collected by filtration, repeatedly rinsed with distilled water until neutral, then dried at 45 °C under vacuum for 36 h and ground for further use [35].

### 2.3. Preparation of PUF and PUF–FRs

The rigid polyurethane foams (PUFs) were fabricated using a one-shot and free-foaming approach, and the specific formulations for the PUFs can be observed in Table 1. To begin, the A-side components, which included polyol, water (blowing agent), silicone oil (foam stabilizer), DBTDL (catalyzer), melamine–formaldehyde resin encapsulated ammonium polyphosphate (MFAPP), and A-lignin, were combined in a 250 mL beaker. This mixture was vigorously stirred at 1000 rpm for 10 min to achieve a uniform dispersion. Subsequently, the B-side components, primarily consisting of PAPI and maintaining a constant NCO/OH ratio (NCO index) of 1.1, were introduced into the beaker. The entire mixture was stirred vigorously at 1200 rpm for 40 s until achieving homogeneity. Following this, the blend was promptly poured into an open plastic mold measuring 250 mm × 250 mm × 200 mm, allowing for vertical expansion of the system. Finally, the samples were put into an oven at 80 °C for 24 h, which could accelerate the polymerization reaction. At last, the PUFs were taken out of the plastic mould, and specimens were cut into standard dimensions for different measurements.

### 2.4. Characterization

The LOI values of samples were measured by a JF-3 oxygen index instrument (Nanjing Jiangning Co., Nanjing, China) at room temperature according to the standard procedure GB/T 2406-1993 [36]. The dimension of the specimens was 100 × 10 × 10 mm^3^.

The vertical burning tests were carried out on a CZF-5 (Nanjing Jiangning Co., Nanjing, China) according to the UL-94 test standards, and the dimension of specimens was 130 × 20 × 20 mm^3^. At least five specimens were tested for each sample, and the average values are reported.

An STA 6000 Thermal Analyser (PerkinElmer, Waltham, MA, USA) was used to investigate the thermal stability of samples. Samples of about 5 mg were heated over 50–800 °C at a heating rate of 10 °C/min under N_2_ at a flow rate of 20 mL/min.

The burning behaviors of samples were tested by an FTT cone calorimeter (Fire Testing Technology, East Grinstead, UK). The specimens of 100 × 100 × 20 mm^3^ were measured with a 35 kW/m^2^ external heat flux.

The micromorphologies of char residues after the cone test and of the PUFs were analyzed by a FEI QUANTA-200 (Eindhoven, The Netherlands) scanning electron microscope, and an energy dispersive X-ray spectrometer (EDX) was utilized for the analysis of the char residue elements.

The apparent density of samples was measured following GB/T6343-2009 standards [37]. Five different specimens of approximately 50 × 50 × 50 mm^3^ for each sample were tested, and the average values are reported.

Thermal conductivity was studied by using a TPS-2500S transient plane source thermal constant analyzer (Hot-Disk, Gothenburg, Sweden). The tests were performed at room temperature by using two samples, which were cut from the same materials with the dimension of 30 × 30 × 10 mm^3^. At least three measurements were obtained for each sample. An average of values for the thermal conductivity is reported for each foam.

The compressive strength of foams was tested with a UTM2503 electronic universal testing machine (SUNS, Zhangzhou, China) according to GB/T8813-2008 standards [38]. The samples of 50 × 50 × 50 mm^3^ were studied at a speed of 5 mm/min. And the relative deformation amount for each measurement was 10%. Each set of specimens was equally divided into three groups of at least five samples each. The first group of samples was placed in a refrigerator at −20 °C for 12 h; the second group of samples was placed in a blast oven at 80 °C for 12 h; and the last group of samples was placed at room temperature (25 °C) for 12 h. Finally, the three groups of samples were removed and immediately tested for compression performance, accompanied by real-time temperature measurement with an infrared temperature gun (at 0 °C, 25 °C, and 60 °C). The average compressive strength values are reported.

The dynamical mechanical analysis (DMA) was carried out by a Q800 DMA instrument (TA Instrument, New Castle, DE, USA) with an amplitude of 25 μm at 1 Hz. The sample size was 50 × 10 × 5 mm^3^. The temperature range tested was from 30 to 250 °C, at a heating rate of 3 °C/min.

The Raman spectra of char residues after the cone test were observed by a SPEX-1403 laser Raman spectrometer (SPEX, Metuchen, NJ, USA) with a 532 nm laser source.

## 3. Results and Discussion

### 3.1. Flame Retardancy of PUFs

To assess the impact of A-lignin and MFAPP on the flame retardancy of PUFs, LOI and UL-94 tests were conducted, and the resulting data are presented in Table 2. In fact, the reference PUF exhibited a pronounced combustibility, receiving no rating in the UL-94 test and registering a mere 18.6% LOI value. When MFAPP alone was mixed in, the LOI value of the PUF/MFAPP_7_ increased to 21.4%, and the form passed a UL-94 V-1 rating with a burning time of 12.18 s. This phenomenon could be attributed to the decomposition of MFAPP into phosphoric acid, metaphosphoric acid, and non-combustible gases like ammonium and water vapor. These byproducts serve to dilute combustible gases, contributing to the observed results [39]. It was obvious that the incorporation of MFAPP and A-lignin at the weight ratio of 4:1 enhanced the flame retardancy, and PUF/MFAPP_5.6_/A-lignin_1.4_ with an LOI value of 22.0% achieved a UL-94 V-0 rating, with the burning time reduced to 8.04 s, which indicated that MFAPP and A-lignin exhibited a cooperative effect on flame retardancy. Upon increasing the incorporation of MFAPP and A-lignin to 15%, the PUF/MFAPP_12_/A-lignin_3_ likewise attained a UL-94 V-0 rating, exhibiting a burning time of 3.44 s. Additionally, the LOI value surged to 23.1%. This enhancement could be attributed to the cooperative interaction between MFAPP and A-lignin, which facilitated the formation of a protective char layer to protect the materials [40]. In comparison, in the work of other researchers, sodium lignosulfonate with APP was used as a flame retardant for improving the fire safety of RPUF, and the LOI value of the material increased merely by 1%, when the flame retardant was added at 30 wt%. In contrast, our results showed that the LOI value of the PUF material was increased by 4.6% when the addition of MFAPP with A-lignin was only 15 wt% [41]. Furthermore, flame-retardant hard segment (HSFR) was used to promote the fire resistance of PUF. When HSFR was added by 16 wt%, the PUF materials just reached a UL-94 V-0 rating [42]. In summary, MFAPP and A-lignin showed positive flame-retardant effects.

### 3.2. Thermal Stability of PUFs

To better study the role of MFAPP and A-lignin in the flame retardancy of PUF composites, the thermal stability of PUFs was researched by thermogravimetric analysis (TGA). The thermogravimetry (TG) and derivative thermogravimetry (DTG) curves of the PUFs under an N_2_ atmosphere are depicted in Figure 1, while the corresponding thermogravimetric analysis data are provided in Table 3.

As shown in Figure 1 and Table 3, the initial degradation temperature (T_5%_) of neat PUF, which was the temperature at 5% mass loss, was 236.4 °C. And there was one degradation stage; during this stage, the maximum thermal-degradation rate (R_max_) was 6.5%·min^−1^, and the temperature at R_max_ (T_max_) was 318.2 °C. The residues at 800 °C of pure PUF were 20.21%. However, with the addition of MFAPP, the T_5%_ of PUF/MFAPP_7_ was decreased a little to 235.7 °C, due to the lower thermal stability of MFAPP. The R_max_ and T_max_ of PUF/MFAPP_7_ were increased to 8.3%·min^−1^ and 320.6 °C, which was owed to the better flame retardancy of MFAPP; the char residues of PUF/MFAPP_7_ were also increased to 25.49% at 800 °C. When A-lignin and MFAPP at the mass ratio of 1:4 were added, the T_5%_ of PUF/MFAPP_5.6_/A-lignin_1.4_ was increased to 239.0 °C, and the R_max_ and T_max_ of PUF/MFAPP_5.6_/A-lignin_1.4_ were increased to 8.2%·min^−1^ and 316.8 °C; and the char residues at 800 °C were increased to 26.66%. Compared to the values for PUF/MFAPP_7_, the T_5%_ was increased, while the R_max_ and T_max_ were decreased. This might be that the A-lignin had a higher thermal stability, and the A-lignin and MFAPP had a cooperative effect; they were combined to form an intumescent flame-retardant system; the higher char residues were attributed to the outstanding catalytic carbonization of the flame-retardant system, and the stable, compact, and thick char layer can restrain the materials from further degradation [43]. With the incorporation of A-lignin and MFAPP increased, the T_5%_ of PUF/MFAPP_12_/A-lignin_3_ was increased to 251.7 °C; the R_max_ and T_max_ of PUF/MFAPP_12_/A-lignin_3_ were decreased to 7.4%·min^−1^ and 315.8 °C; and the char residues at 800 °C were increased to 34.81%. Based on these results, A-lignin and MFAPP at the weight ratio of 4:1 could accelerate the formation of thermal insulative residues and increase char-forming ability [41].

### 3.3. Combustion Behavior of PUFs

Cone calorimetric measurement is an ideal method to evaluate the combustion behavior of polymers. The flammability of samples was interpreted by some characteristic parameters, including time to ignition (TTI), peak of heat release rate (PHRR), total heat release (THR), smoke production rate (SPR), total smoke production (TSP), and char yield. To further study the fire behavior of PUFs, pure PUF, PUF/MFAPP_7_, PUF/MFAPP_5.6_/A-lignin_1.4_, and PUF/MFAPP_12_/A-lignin_3_ were characterized by the cone test. And the related data and curves are listed in Table 4 and Figure 2.

The TTI is an important flame-related parameter for polymers. From Table 4, the mixture of MFAPP and A-lignin increased the TTI from 2 s to 4 s, which indicated that the flame retardant slows down burning. Figure 3a shows that pure PUF had two heat release peaks at 20 s and 41s; the peak values were 272.8 and 288.6 kW/m^2^. The first peak corresponds to the combustion of materials leading to char formation, while the subsequent peak pertains to the degradation of the char layer [44]. When MFAPP was added into the composite, the HRR curve had one heat release peak, whose value decreased to 267.0 kW/m^2^ at 21 s. This result was caused by the MFAPP decomposing into polyphosphoric acid, which binds the char residue and crosslinks with polyhydric alcohols, and by the formation of compact and stable char residues, which can protect the inner materials [45,46]. Meanwhile, ammonia gas generated during the combustion could absorb heat to reduce the temperature of the material and dilute the combustible gas to enhance the flame retardancy [47]. With the addition of MFAPP and A-lignin, the foams also had one PHRR during the combustion; the PHRR of PUF/MFAPP_5.6_/A-lignin_1.4_ was 261.7 kW/m^2^ at 19 s, while the PHRR of PUF/MFAPP_12_/A-lignin_3_ was 255.2 kW/m^2^ at 19 s, which is because the A-lignin promotes combustion on the surface of materials and the formation of more char layer [48]. Figure 4b shows the THR curves; it shows that the THR of pure PUF was 15.4 MJ/m^2^, and the THR of PUF/MFAPP_7_ was 12.3 MJ/m^2^. When MFAPP and A-lignin were present, the THR values were reduced to 11.1 (PUF/MFAPP_5.6_/A-lignin_1.4_) and 10.0 MJ/m^2^ (PUF/MFAPP_12_/A-lignin_3_). As a result, it can be indicated that the inclusion of MFAPP and A-lignin had a better effect on the flame retardancy of PUF.

The SPR and TSP curves are presented in Figure 3c,d. Seen in the curves and Table 4, the peak SPR value of pure PUF was 0.10 m^2^/s with a TSP value of 3.05 m^2^. When MFAPP was added individually, the PSPR and TSP increased to 0.14 m^2^/s and 3.65 m^2^. The reason is that a lot of smoke was released immediately because of the degradation that was promoted by MFAPP. With the incorporation of MFAPP and A-lignin, the PSPR value of PUF/MFAPP_5.6_/A-lignin_1.4_ decreased to 0.13 m^2^/s with the TSP value of 3.37 m^2^. Upon continuing to increase the amount of flame-retardant system, the PHRR value was reduced to 0.08 m^2^/s, which was 20% lower than for pure PUF. And the TSP value was also decreased to 2.89 m^2^, which was caused by the protective effect of the formed char layer and the inhibition effect created by MFAPP and A-lignin.

As seen in Table 4, pure PUF burns rapidly with a final char residue value of 14.5%. Compared to pure PUF, the char residue value of PUF/MFAPP_7_ and PUF/MFAPP_5.6_/A-lignin_1.4_ increased to 16.8% and 18.8%, and the residue value of PUF/MFAPP_12_/A-lignin_3_ was 21.9%, which was 51% higher than for pure PUF. This result is consistent with the test results obtained by thermogravimetric analysis.

### 3.4. Physical and Mechanical Properties of PUFs

Density is a critical parameter that significantly affects the physical properties of PUFs. It is closely connected with material properties such as dimensional stability, thermal insulation, and various mechanical characteristics [49]. The density of PUFs was tested and is presented in Table 5. It is shown that the density of pure PUF was 26.7 kg/m^3^. When 7 wt% MFAPP was added, the density increased to 29.9 kg/m^3^. With the incorporation of 7 wt% MFAPP and A-lignin, the value of the density was enhanced to 32.2 kg/m^3^. Along with the increasing incorporation of flame-retardant, the density gradually increased to 35.4 kg/m^3^.

Thermal conductivity is an important physical property of polyurethane foam, which is closely related to the application of polyurethane foam in the field of building insulation [50]. The values of thermal conductivity were determined at room temperature, and the corresponding data are shown in Table 5. The thermal conductivity value of pure PUF is 29.63 mW/(m·K). When MFAPP was added, the thermal conductivity value of PUF/MFAPP_7_ increased to 30.85 mW/(m·K). With the addition of MFAPP and A-lignin, the thermal conductivity value was enhanced to 31.93 mW/(m·K) (PUF/MFAPP_5.6_/A-lignin_1.4_) and 32.68 mW/(m·K) (PUF/MFAPP_12_/A-lignin_3_). The result might be owed to the change in the foam structure, which was caused by the incorporation of flame retardant, and the incorporation had an effect on the density and porosity of foams, which could be proved in Figure 3. The incorporation of flame retardant decreased the cell size. As depicted in Figure 3a, the morphology of pure PUF featured cells of a relatively regular, closed polyhedral structure, with pore sizes averaging around 400 μm. Incorporation of 7 wt% MFAPP resulted in PUF/MFAPP_7_ possessing cells of irregular shapes and reduced pore sizes, varying from 200 to 500 μm, as illustrated in Figure 3b. With the concurrent addition of MFAPP and A-lignin, the pore sizes in the PUFs ranged from 200 to 400 μm, and the cells exhibited a more uniform shape, as demonstrated in Figure 3c,d.

The compressive strength data are presented in Table 5. At room temperature (25 °C), the compressive strength value of pure PUF was 132.4 kPa. When MFAPP was added into the composite, the compression performance of the foam containing MFAPP was slightly worse than that of the pure PUF, which was only 110.0 kPa. That is because the MFAPP and the PU matrix had a bad interfacial adhesion, which led to slippage among these two components. The addition of MFAPP damaged the cell structure of the PUF, resulting in an inhomogeneous cell structure and decreasing the compressive strength, which is shown in Figure 3b. Nevertheless, the compressive strength of both PUF/MFAPP_5.6_/A-lignin_1.4_ and PUF/MFAPP_12_/A-lignin_3_ increases with the addition of A-lignin. In PUF/MFAPP_5.6_/A-lignin_1.4_, the value was 150.2 kPa, which was higher than that of pure PUF by 13.4%. When the addition amount of the flame retardant was 15 wt%, PUF/MFAPP_12_/A-lignin_3_ showed a better compression performance with the value of 160.5 kPa. For a comparison of specific values (dividing the compression strength by the density), the PUF/MFAPP_7_ was 3.67; with the incorporation of A-lignin and MFAPP, the value increased to 4.63 (PUF/MFAPP_5.6_/A-lignin_1.4_) and 4.53 (PUF/MFAPP_12_/A-lignin_3_). It suggested that A-lignin could be a potential mechanical enhancement agent to solve the negative effects of MFAPP. At 0 °C and 60 °C, the compressive strength of the PUFs had not changed greatly, which showed that the A-lignin flame-retardant PUFs had a good mechanical property at both high or low temperatures.

The storage modulus is an extremely important factor in reflecting the stiffness of a polymer. Therefore, DMA was utilized to examine the stiffness of PUFs. Figure 4 presents the variations of the storage modulus as the functions of temperature.

As revealed in Figure 4, when MFAPP was added into the composite, the storage modulus of PUF/MFAPP_7_ was significantly reduced. With the incorporation of A-lignin and MFAPP, the storage modulus of PUF/MFAPP_5.6_/A-lignin_1.4_ was obviously improved. As the amount of flame retardant added continuously increased, the storage modulus of PUF material was higher than that of pure PUF, which indicated that the flame-retardant PUF material had an excellent stiffness. The results of the DMA were in unison with the mechanical properties test results of the PUFs.

### 3.5. Char Residue Analysis of PUFs

To further explore the flame-retardant mode of action of the co-incorporated A-lignin and MFAPP in PUF, SEM was employed to examine the char residues after CCT. The digital micrographs of the char residues and the micromorphologies of the outer surface char residues are presented in Figure 5.

Shown in Figure 5, there were many holes on the outer char residues of pure PUF, and the char residues were sparse and cracked. Therefore, they could not prevent the materials from burning as a shield. On the other hand, the inner structure of the char residues was noncompact and empty, which indicated poor flame retardancy. When MFAPP was added, a continuous and expanded char residue was formed. However, the char residues became more continuous, intact, and thick after being added into with MFAPP and A-lignin. The addition of alkali lignin makes the char layer more expansive as well as improving the firmness of the char layer. The expansion char layer could slow down the heat/O_2_/flammable gas transfer during the combustion process, which could prevent the underlying polymeric substrate from further burning [51]. This could obviously improve the flame retardancy of PUFs.

In the element distribution mappings of EDX, the spatial distributions of EDX, N, and P elements in PUF/MFAPP_12_/A-lignin_3_ char residues after the cone calorimetric test can be obviously observed in Figure 6. Seen in Figure 6, the elements of P and N still remain in the char residues after burning and reveal an even distribution. And the char layer could be used as a shield to protect the underlying foam.

The Raman spectra of char residues after CCT were observed by a SPEX-1403 laser Raman spectrometer. As seen in Figure 7, there were two absorption peaks at around 1350 and 1595 cm^−1^ of the samples, corresponding to the D and G bands. The intensity ratio of the D to G peak was denoted as I_D_/I_G_, corresponding to the height ratio. The lower I_D_/I_G_ value indicates a higher graphitization degree, which is corresponding to the better flame retardancy. The I_D_/I_G_ value of PUF was 1.52, which was the highest I_D_/I_G_ value. It is noteworthy that the I_D_/I_G_ values of PUF/MFAPP_7_, PUF/MFAPP_5.6_/A-lignin_1.4_, and PUF/MFAPP_12_/A-lignin_3_ were 1.19, 1.17, and 1.10, respectively, implying that the incorporation of MFAPP and A-lignin can endow a better graphitization degree upon PUF. PUF/MFAPP_12_/A-lignin_3_ showed the lowest I_D_/I_G_ ratio, which indicated that the addition of MFAPP and A-lignin with a suitable ratio could promote the production of better protective char layers to endow PUF with better fire safety.

In short, the flame-retardant mode of action of PUF/MFAPP_12_/A-lignin_3_ is demonstrated in Figure 1. Upon ignition, PUF/MFAPP_12_/A-lignin_3_ underwent a process resulting in the formation of a dense and durable residual layer enriched with phosphorus and nitrogen derivatives. This layer demonstrated a barrier effect, effectively impeding the transfer of heat, oxygen, and other thermal degradation byproducts to the surface. Meanwhile, ammonia gas generated during the combustion could absorb heat to reduce the temperature of the material and dilute combustible gas to enhance flame retardancy. Through comprehensive analyses in both the condensed phase and the gas phase, it became evident that the addition of MFAPP and A-lignin significantly enhances the fire safety performance of PUF.

## 4. Conclusions

In this work, A-lignin was used as a synergistic agent and incorporated with MFAPP into rigid polyurethane foam to improve the flame retardancy of PUF/MFAPP. The LOI value of PUF/MFAPP_12_/A-lignin_3_ was increased to 23.1%, with the mass ratio of MFAPP to A-lignin of 4:1, and PUF/MFAPP_12_/A-lignin_3_ also achieved a UL-94 V-0 rating. MFAPP and A-lignin had an effect on the density and compressive strength of PUF–FRs: the density value of PUF/MFAPP_12_/A-lignin_3_ was increased to 35.4 kg/m^3^, and the compression performance was higher than that of the pure PUF by 13.4% and increased to 150.2 kPa. And the thermal conductivity value was enhanced to 32.68 mW/(m·K). Furthermore, the initial decomposition temperature (T_5%_) of PUF/MFAPP_12_/A-lignin_3_ was increased to 251.7 °C. And the char residues were also increased to 21.9% at 800 °C, which was higher than that of the pure PUF. Compared to those of the pure PUF, the values of PHRR, THR, SPR, and TSP were decreased, which was caused by the intact, thick, and continuous char layer with an abundance of nitrogen and phosphorus elements. This work provided an alternative for flame retardant modification research by using environmentally friendly lignin, which is of great significance for the comprehensive utilization of lignin.

## Data Availability

The data presented in this study are available on request from the corresponding author.

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
