# Peer review of "Bio-Based Alkali Lignin Cooperative Systems for Improving the Flame Retardant and Mechanical Properties of Rigid Polyurethane Foam"

_polymers, 2023, doi:10.3390/polym15244709_

Round 1

Reviewer 1 Report

Comments and Suggestions for Authors

The manuscript titled “Bio-based Alkali Lignin Synergistic System for Improving the Flame Retardant and Mechanical Properties of Rigid Polyurethane Foam” by Li et al. describes the use of a-lignin as flame retardant agent to be incorporated into polyurethane foams. The presented results are of interest, I have few minor comments:

1) The Authors state that the incorporation of the flame retardant decreases the PUF cell size. Did they try to quantify this effect by systematically analysing SEM micrographs? What do they mean by “large particle size of MFAPP”? Please provide more detailed information on this aspect.

2) Line 95: define “PAPI”

3) Table 1: how is the mass content defined? The sum of all the components for each formulation does not lead to 100%...

4) Line 160: define “TG” and “DTG” curves

5) Line 256 and 300: I suggest substituting “SEM photos” with “SEM micrographs”

6) Line 70, 164, 175, 199, 200, 203, 235: check sentence grammar

Comments on the Quality of English Language

English language is overall fine. Authors must review some sentences as suggested in "Comments for Authors" section.

Reviewer 2 Report

Comments and Suggestions for Authors

The article analysis flame retardant and mechanical properties of polyurethane foam filled with modified alkali lignin. The idea is very interesting but I do not think that it is completely fulfilled. My remarks are indicated below:

1. Introduction section needs to be improved with more in depth analysis of literature about the pure fillers and modified fillers used to enhance the fire resistance of polyurethane foams.

2. Please correct the dimensions of viscosity in Materials section.

3. The common mixing time of polyurethane components is 10 seconds. 40 seconds is too much as it indicates that the amount of selected catalysts are too low. Is there any reason to add lower amount of catalysts or not to add it? What about blowing agent used? There is no information given about this raw material in Materials section.

4. I would suggest proofreading the article as it seems there are few grammar errors.

5. The study misses the discussion with other authors' works.

6. Please correct the dimensions of HRR and THR in y axes in Figure 2.

7. the study is missing DMA results and mechanical properties at different temperatures results.

Round 2

Reviewer 2 Report

Comments and Suggestions for Authors

Authors have taken into consideration all my remarks.

Author Response

Thanks for your recognition.